# Barriers and Facilitators to Timely Diagnosis of Tuberculosis in Children and Adolescents in Karachi, Pakistan

**DOI:** 10.3390/ijerph22101477

**Published:** 2025-09-24

**Authors:** Sara Ahmad, Maria Jaswal, Amyn Abdul Malik, Maria Omar, Iraj Batool, Ammad Fahim, Hannah N. Gilbert, Carole D. Mitnick, Farhana Amanullah, Courtney M. Yuen

**Affiliations:** 1Department of Global Health and Social Medicine, Harvard Medical School, Boston, MA 02115, USA; hannah_gilbert@hms.harvard.edu (H.N.G.); carole_mitnick@hms.harvard.edu (C.D.M.); courtney_yuen@hms.harvard.edu (C.M.Y.); 2Baitussalam Welfare Trust, Karachi 75500, Pakistan; mariaraufjaswal@gmail.com; 3UT Southwestern Medical Center, O’Donnell School of Public Health, Dallas, TX 75390, USA; amyn.malik@ird.global; 4Interactive Research and Development (IRD) Global, TB Program, Singapore 048581, Singapore; farhana.amanullah@childrens.harvard.edu; 5Indus Hospital and Health Network, Karachi 75190, Pakistan; maria.omar@tih.org.pk (M.O.); iraj.batool@tih.org.pk (I.B.); ammad.fahim@tih.org.pk (A.F.)

**Keywords:** tuberculosis, adolescent, child, delayed diagnosis

## Abstract

Background: The diagnosis of tuberculosis (TB) in children and adolescents is often delayed. We conducted a study to understand the barriers and facilitators to the diagnosis of TB in children and adolescents in a not-for-profit private hospital in Karachi, Pakistan. Methods: We conducted a convergent mixed-methods study comprising quantitative surveys with caregivers of 100 TB patients < 18 years old and 40 semi-structured interviews with caregivers and healthcare providers. Results: Among TB patients whose caregivers were surveyed, 82% were adolescents 10–17 years old. Caregivers reported a median of 91 days (IQR 58–160) between symptom onset and treatment initiation. Time was divided relatively evenly between symptom onset and the first visit to a healthcare provider (median 73, IQR 42–130 days), and between this visit and TB diagnosis (median 65 days, IQR 30–114). While 69% of caregivers initially visited general physicians, many felt that these general physicians did not provide satisfactory healthcare. Caregivers mentioned financial constraints as a major barrier affecting all stages of the journey to diagnosis and treatment. Conclusions: Interventions that overcome financial barriers and strategies that enhance the capacity of private sector general physicians are necessary to reduce delays in TB diagnosis and treatment initiation for children and adolescents.

## 1. Introduction

Of the approximately 10 million people who contract active tuberculosis (TB) each year, one in every six is a child (0–9 years old) or adolescent (10–19 years old) [1,2]. Despite being both curable and preventable, TB remains a top-ten killer of children under five years old globally [3] and one of the top three causes of death among adolescents [4]. Nearly 700,000 children and young adolescents (0–14 years) with TB were reported to the World Health Organization (WHO) in 2023; although this was the largest number reported annually to date, WHO estimated that almost half of children with TB still had not been diagnosed and treated [5]. Challenges with TB diagnosis in children include difficulty producing sputum, paucibacillary disease that is difficult for bacteriologic testing to detect, and non-specific symptoms [6]. Adolescents with TB also experience unique challenges with diagnosis and treatment [1], such as living at boarding schools where there is limited TB care access [7]. In Pakistan, limited awareness and information on child and adolescent TB diagnosis among providers, as well as the absence of services, further compromises TB detection [8]. Pakistan has the fifth highest TB burden globally with an estimated incidence of 277 per 100,000 people in 2023 [2], and the Sindh region where the city of Karachi is located is estimated to have the highest TB incidence in the country [9].

Timely diagnosis of TB and initiation of appropriate treatment play a significant role in reducing disease transmission and improving patient outcomes [10]. While many studies have quantified delays in TB treatment initiation in adults and the reasons for these delays [11,12], fewer studies have done so for children and adolescents. The conclusions from adult-focused studies cannot be simply extrapolated to children and adolescents given challenges in diagnosing pediatric TB and differences in how children and adolescents access health care compared to adults. In a systematic review of 124 studies published during 2008–2018 and evaluating delays in TB treatment in countries with high TB burdens, only 5 focused on children or adolescents [12]. Studies quantifying time between symptom onset and diagnosis among children and adolescents with TB have largely come from India and China, where delays ranging from 2 to 8 weeks have been driven by inadequate knowledge about tuberculosis among caregivers and multiple referrals to different facilities before receiving proper care [13,14,15].

Globally, it is important to add to the sparse literature that seeks to quantify and understand delays in treatment initiation experienced by children and adolescents with TB. This knowledge is necessary for countries to make improvements to TB services to ensure prompt treatment for this vulnerable population. No study on this topic has taken place in Pakistan. We therefore conducted the present study to identify barriers and facilitators to timely diagnosis of TB in children and adolescents. This will inform national policies for enhancing the care of TB-affected children and adolescents.

## 2. Methods

We conducted a convergent mixed-methods study to quantify delays in diagnosis and identify barriers and facilitators that affect timely TB diagnosis and treatment of children and adolescents in a hospital in Karachi, Pakistan.

### 2.1. Setting

In Karachi, healthcare is delivered in a variety of places, including public hospitals, as well as private clinics and hospitals. Private clinics are typically run by general practitioners (GPs) and are usually not affiliated with hospitals, whereas private hospitals are run by for-profit companies, non-governmental organizations, and faith-based organizations. The present study was conducted in the TB clinic of a private, non-profit hospital (referred to as Hospital A) in Karachi that offers services completely free of cost to all patients. The hospital mostly serves economically disadvantaged communities. Additionally, the hospital’s pediatric TB program registers 150 to 200 children and adolescents with TB every month, who are often referred with symptoms suggestive of TB from other health facilities. All forms of TB, including drug-resistant TB, extrapulmonary TB, and TB meningitis are diagnosed and treated. Diagnosis and treatment at Hospital A’s TB clinic are done per national TB program guidelines.

### 2.2. Participants

For the quantitative component of the study, we administered surveys to adult primary caregivers of 100 consecutively registered children and adolescents under 18 years of age, who were registered as TB patients at Hospital A’s TB clinic between April–December 2023, and who were receiving TB treatment at the time of the survey. For the qualitative component, we conducted 40 individual, semi-structured interviews. Participants included 30 caregivers selected from among caregivers surveyed for the quantitative component. We used purposeful sampling in order to ensure variation according to patient ages. In addition, we interviewed 10 healthcare providers who were employed at Hospital A between April–December 2023, and who were involved with the TB program. The healthcare providers included 2 physicians, 1 nurse, 1 clinic manager, 1 laboratory manager, and 5 community healthcare workers.

### 2.3. Data Collection

For the quantitative component, we collected demographic and clinical information from electronic medical records, including date of TB diagnosis and date of TB treatment initiation. The lead author, a native Urdu speaker, verbally administered a 30 min survey to caregivers at the TB clinic or over the phone in the Urdu. We adapted an established survey instrument [16] to capture details about the healthcare-seeking journey, including date of symptom onset, date of first healthcare facility visit, and what type of healthcare providers they chose to visit.

Qualitative interviews lasting 60–90 min were performed by the lead author in Urdu. Participants were mostly interviewed in the TB clinic; a few caregivers were interviewed at their homes. Interview guides for caregivers focused on their journey from the child’s symptom onset to initiation of TB treatment, including the role of family members and the reasons for healthcare seeking decisions. Interview guides for healthcare providers focused on factors at the hospital that could affect the time taken for children and adolescents to be diagnosed with TB and started on treatment, and their opinions of patients’ experiences during this process. Interviews were audio-recorded and transcribed into English for analysis by the bilingual lead author.

## 3. Analysis

For quantitative analysis, we calculated the time (in days) from symptom onset to diagnosis, subdivided into three intervals: (1) from onset of symptoms until the first visit to a healthcare provider, (2) from the first visit to a healthcare provider until the child was diagnosed with TB, and (3) between the TB diagnosis and initiation of treatment. We choose not to use the common terms of “patient delay,” “health system delay,” and “pre-treatment delay” to avoid ascribing blame [12]. We analyzed these values descriptively.

For qualitative analysis, we used a content analysis approach [17]. After reviewing all interview transcripts, SA open-coded a subset of transcripts. The open codes were reviewed with HNG and CMY and used to develop a draft codebook, which was piloted and revised. Dedoose v9.0.107 (SocioCultural Research Consultants LLC, Los Angeles, CA, USA) qualitative data management software was then used to code the entire dataset. We used an inductive approach to examine the coded data, by identifying an initial set of emergent themes which were described and illustrated with quotes from the data. Using an iterative process, these initial themes were revised into a set of final explanatory themes [18].

For integration, we used the qualitative results to explain the reasons behind the delays measured in the quantitative component using the method of joint display.

## 4. Results

### 4.1. Quantitative Results

Among the 100 TB patients whose caregivers were surveyed, 77% were female and most fell into the adolescent age groups (Table 1). Parents of the patients comprised 80% of the caregivers who were surveyed.

Caregivers reported a median of 91 days (interquartile range [IQR] 58–160) between onset of symptom(s) and treatment initiation (Table 2, Figure 1). Similar amounts of time were reported from symptom onset to first healthcare visit (median 73, IQR 42–130 days) and from first healthcare visit to diagnosis (median 65, IQR 30–114 days). After diagnosis, treatment was initiated after a median of 1 day (IQR 0–5). Patients made a median of 4 (IQR 3–6) visits to healthcare providers en route to diagnosis. When asked about the type of healthcare provider they went to first, 69% chose private practice, while 28% went to hospitals directly, 2% went to a laboratory or pharmacy, and 1% went to a spiritual healer.

### 4.2. Qualitative Results and Integration

Among caregivers who were interviewed, 27 (90%) were parents, of whom 17 (57%) were female (Table 1). While we attempted to recruit equal numbers of caregivers of patients in different age groups, the small number of children in the quantitative sample meant that only 10 (33%) interviews were caregivers of children < 10 years old. Study participants reported several barriers and facilitators that affected the time taken for different stages of the diagnostic and treatment journey, which can be categorized under three broad categories. Aspects of the patients’ environment affected all stages of the journey. Involvement of general physicians affected the time between the first visit to a healthcare provider and treatment initiation. Attributes of Hospital A affected all stages of the journey, although different processes and systems affected different stages. Figure 1 presents a joint display showing which stages of the journey were impacted by each specific category of barriers/facilitators.

Patient’s environment

Relationships in home and community

Gender norms

Caregivers shared that their families identified with conventional gender-based roles, such as women staying at home and taking care of the household chores and their families, while the men worked outside the home, earned a living, and took care of the responsibilities that entail leaving the house. In most cases, women caregivers were not allowed or were unable to work, making them economically dependent on their partners. This led to frustration for both partners, which in a few cases culminated in violence afflicted by the husband on his wife and children. These gender dynamics affected women caregivers’ wellbeing and ability to seek healthcare for their sick child.

“Since my child is sick, I cannot earn money, which means [my husband] is forced to go and work and pay all the bills and rent. That makes him angry and he has made our lives miserable—he yells at us almost every day, as well as hits us. How can I ask him to give me money to buy my child medicine or to take her to the doctor? I am too scared.” (Caregiver ID 98, female)

Healthcare providers felt that in families where men took some responsibility for the sick child’s caregiving, the child would get better care. They explained that men decided in most cases how money was spent, which health facility to use, and how seriously to take the child’s illness. Men were also perceived to be more confident taking public transport, attending facilities far from home, and talking to healthcare providers.

“We have seen that when the fathers take interest in their child’s treatment, even if the mother does not, such children tend to do better. But we have seen the most trouble with the fathers. Even when we counsel them about taking care of the mothers, they do not seem to care about the women… They do not bring in their children for follow-ups.” (Healthcare provider ID 105)

Assistance with childcare

Caregivers reported that help from extended family or friends to look after other children at home was a vital facilitator to seeking care for the sick child. This was particularly important for single mothers and for families where there were multiple young children. It was shared that childcare support enabled female caregivers to take their children to healthcare facilities multiple times without risking loss of income by the family breadwinner.

“My family helped in taking care of my other children. Everyone in my family helped, my father lives with us so he provided a lot of support. Since the family lives with me I did not have to worry about my other kids.” (Caregiver ID 87, male)

Competing social priorities

Some caregivers reported that although they knew that their child needed to see a doctor, unavoidable circumstances led to a significant amount of time passing before seeking care. These included marriage, death, or other illness in the family, and both parents having work obligations.

“I could not go back to Hospital A as my father in the village fell sick and I had to go with the family to the village. Soon after we got to the village, he died and we had to stay there for a few months. The doctor was very surprised and angry at us for taking this long to come back. She said we harmed our daughter’s health by causing this delay, and that we should have come back much sooner. I told her that I was helpless, that my father was dying” (Caregiver ID 06, female)

Substance use

Some caregivers reported that having a partner who uses drugs affects the caregivers’ health-seeking for their children. They said that these partners spend the family’s savings on procuring drugs and are unable to seek jobs because of the addiction. This leads to financial constraints on the family and prevents the caregivers from bringing the sick child to a healthcare facility.

“My husband takes a drug called ‘ICE’, because of which he does not work, nor takes any interest in helping me bring the patient to the hospital.” (Caregiver ID 84, female)

Direct referral or accompaniment

Several caregivers reported that a person in their family or community was more aware of TB symptoms and the care that is needed for it, facilitating their care-seeking. In some cases, they first found out about Hospital A offering free and quality care from this person. Caregivers also shared how their journey to health facilities was facilitated by physically being accompanied by someone they trusted. This helped the caregivers to not only reach the facilities, but also to navigate their way inside them. In most cases, this support was provided by people who had more formal education than the caregivers.

“My office colleague was very kind and he accompanied me to Hospital A with my wife and child. This is because I had never been to Hospital A before, and he had. He was very helpful in taking us to the correct departments and getting everything done in the same day.” (Caregiver ID 82, male)

2.Financial constraints

Fees for consultations, tests, and medicines

Challenges due to the cost of seeking care were reported by almost all caregivers. They reported spending money on transportation to the health facilities, doctors’ fees, laboratory and radiological tests, and medicines. These expenses were burdensome for most families, who often borrowed money from employers, family, and friends. While this enabled caregivers to seek care for their children, they shared that the longer-term repercussions for incurring debts negatively impacted their mental health, financial savings and family relationships. Some caregivers were deeply financially indebted to private hospitals where their children were admitted for weeks before receiving a TB diagnosis.

“[My parents] already had taken a loan for my brother’s treatment from people in the neighborhood, which they are pressurized to repay… My dad works as a plumber and loses out on work whenever they take my sister to the hospital. It is a miracle they can feed my siblings since the work is not stable and they are already in debt. If we had some money, my siblings could have gotten better treatment.” (Caregiver ID 49, female)

Consequences from taking time off from work

Caregivers shared that the breadwinners in patients’ families were often daily wage workers who would lose a day of wages if they spent the day bringing their child for healthcare. Lost work meant families wouldn’t have money to buy food that day. Caregivers working office jobs could also lose pay if they did not show up at work. Multiple visits to healthcare facilities compounded these economic risks, with some caregivers worrying that taking too many days off from work could cause permanent loss of employment. Families feared job loss, particularly in times of economic recession.

“I also worry about missing so many days at work because it is a private job, and they are downsizing these days. What if my name is also on that list? We live on rent, it would be hard to pay it if I won’t have work.” (Caregiver ID 100, male)

3.Geographic accessibility to services

Distance and transportation to healthcare facility

Caregivers stated that distances between their homes and healthcare facilities affect their healthcare-seeking decisions. Public transport to facilities that were far away was viewed by caregivers as unaffordable, time-consuming, confusing, unsafe, or uncomfortable. In some cases, caregivers chose to spend the night at the hospital to save the cost on transportation. Private vehicles, while more convenient, required a male partner to drive them. Several caregivers mentioned that they preferred borrowing a motorcycle or bicycle from friends or family.

“I borrowed a motorbike from a friend in the neighborhood… He is a very good friend—but can only give the bike when he is home and not at work. He does not even take money for the fuel. He knows we hardly have enough money in our house for food, let alone fuel for the bike.”(Caregiver ID 90, male)

Lack of community screening services

Healthcare providers shared that community screenings are a useful and effective way to be able to identify TB early in children. Previously there were programs for community screenings, household contact screenings for families of TB patients, and awareness campaigns. Those programs were stopped because of funding constraints, and providers noted that limited resources have led to a decline in the number of children being diagnosed with TB.

“We have such limited resources that we cannot go and do mass screening campaigns, even though TB is rampant in our communities.”(Healthcare provider ID 101)

4.Knowledge

Prior knowledge and experiences

Some caregivers shared that prior knowledge of TB helped them recognize their child’s symptoms and access care sooner. This was particularly common in families where other members had experienced TB disease before.

“My husband has had TB before, when he was younger. So when our daughter was admitted in [a semi-private hospital] and the doctors would not do anything except give her drips even when she kept getting sicker, he became frustrated. He had her discharged and brought her to the Hospital A, as he had a feeling she might have TB and he knew Hospital A provides TB treatment.” (Caregiver ID 05, female)

Higher literacy

Caregivers noted that family members who were relatively more educated found it easier to navigate healthcare facilities and speak to healthcare providers. Some caregivers mentioned that they depended on the family member who was educated to make the key decisions for anyone in the family who needed to seek healthcare services.

“I got my sister’s tests done [ at a tertiary public hospital], and she was diagnosed with TB of the lungs… Being educated, I had to assume the responsibility as I understand these things quicker than my parents. It is too much responsibility; I get into fights with everyone and have become very irritable. It is very frustrating, but my husband supports me which I appreciate.” (Caregiver ID 49, female)

II.Involvement of general practitioners

Caregivers felt that GPs had insufficient training and therefore did not ask caregivers to get their children tested for TB. In some cases, caregivers reported that GPs overprescribed antibiotics, misdiagnosed the disease (commonly with typhoid, pneumonia or malaria), or had the tests done and yet could not diagnose anything concrete. When TB was diagnosed, GPs would not treat the patients themselves and often referred them to hospitals instead. Several caregivers and healthcare providers suggested that GPs were not able to provide satisfactory healthcare to the patients because they are not qualified or they want to retain the patients for themselves for monetary gain. Patients spent a significant amount of time in the care of GPs, which in some cases led to disease progression and worsening of the patient.

“The next day I took her to the local doctor in the neighborhood called [Dr X]. He checked her fever and examined her left side, where she had been feeling pain when she slept. He gave her medicines for three days, but she did not feel better. So we went back to him, and he gave her many painkillers (around 9–10). That made her feel better. He did not prescribe any tests at all. Then Eid came, and as she was not totally well (weakness, fever and lack of appetite were still there), it felt like the medicines had killed her appetite. So my sister told me to go to another doctor, who is an MBBS doctor. His name was [Dr Y]. He was shocked to hear that she had been having a fever for 15–16 days, and as we did not have Dr X’s [prescription] with us, he gave her new medicines.” (Caregiver ID 01, female)

III.Characteristics of Hospital A

5.Processes

Challenges for patients to complete the diagnostic process

Healthcare providers and caregivers reported that long lines and waiting times can be cumbersome for patients. Patients must stand in a line to get a token, which entitles them to a basic consultation from a doctor. If the tokens finish before the patient gets one, they have to come back later to stand in line again. Several caregivers shared that they had to make two visits to the hospital to make the process easier for their families—the first time to get the token and the second time to bring the sick child.

Patients also faced challenges after obtaining an appointment. Caregivers reported that they had to make several visits to Hospital A before TB diagnosis and treatment initiation, including visits for getting a token to make an appointment, getting tests done, and collecting test results. The appointments were quite spaced out, delaying diagnosis and causing stress for patients and their families. Healthcare providers were aware of the difficulty patients faced in navigating care within the hospital:

“Some patients also can’t navigate the hospital as they are mostly uneducated. I think it would be very helpful to the patients if there are dedicated staff at the hospital who can guide patients through the entire system. I know patients who stood in line for hours only to find out that they had been standing in the wrong line the whole time.”(Healthcare provider ID 102)

Diagnostic testing procedures

Healthcare providers noted both barriers and facilitators to diagnostic testing at the hospital. TB diagnostic testing can be performed on sputum or stool for children, but each has challenges. As it is difficult for children to produce sputum, one healthcare provider suggested that videos with instructions should be shown in the clinic to collect better quality samples. This way patients would avoid sample rejection and a return visit, which wastes time. On the other hand, since stool collection is dependent on the child passing stool, caregivers choose to collect the sample at home and make a separate visit to the hospital to submit it. Once samples were obtained, healthcare providers reported that the TB diagnostic process was efficient. Facilitators included perceived dedication of the laboratory personnel and protocols for prioritizing TB testing.

“Our laboratory staff is dedicated to serving the patients to the best of their capabilities. We try to perform the tests ordered by the TB clinic doctors as soon as possible, even if it is stool GeneXpert, which most other laboratories avoid doing. If the sputum content is too little for all tests, we give preference to GeneXpert so that the doctors have some basis to diagnose the patient and time is not wasted.”(Healthcare provider ID 110)

Organization of service delivery

Collaboration between hospital departments was identified as a facilitator of timely TB diagnosis. Healthcare providers shared that conducting TB screenings in outpatient departments of Hospital A has helped reduce delays in diagnosis and treatment initiation. Both caregivers and healthcare providers mentioned that hospital staff contacted patients by phone to notify them about positive TB diagnostic test results and upcoming appointments, a process which involves linkages between different departments in order to run smoothly. Healthcare providers mentioned that collaboration between the TB clinic and other hospital departments helps distribute the patient load; however, more trainings are needed to make the diagnostic services faster at the TB clinic.

Several healthcare providers also reported that the TB clinic was understaffed due to lack of funds, and that having more doctors would reduce wait times for appointments and hence delays in diagnosis. One provider also mentioned difficult dynamics in the relationship between the Provincial TB Control Program (PTP) and Hospital A’s TB clinic. Challenges in funding salaries of employees at the clinic (which partially comes from the PTP) and ensuring smooth functioning of the X-ray machine led to challenges in efficient service delivery.

“PTP says that this is the maximum they can provide, if more staff is needed then Hospital A will have to do that. And Hospital A does cover salaries of housekeeping and mine, but it is still not enough because our load is so high. In addition, since our operations are in a hybrid mode—this means that PTP bought us the X-ray machine, but Hospital A had to buy the license and make sure to make it work. And this tussle is not ending.”(Healthcare provider ID 103)

6.Infrastructure

Data systems

Healthcare providers referred to the electronic medical record system at Hospital A as a facilitator of timely diagnosis and treatment initiation. The electronic system ensures that all details of the patients’ previous visits to the hospital are readily available for healthcare providers, enabling them to work efficiently and allowing quick referrals between departments.

“Patients’ records are always available at Hospital A, while in the other hospitals, the patient’s record is given to the patient—this leads to them losing the records or sometimes they forget to bring them when re-visiting the doctors.”(Healthcare provider ID 110)

Lack of physical infrastructure

Healthcare providers discussed the lack of isolation rooms in the hospital which prevents the hospital from admitting patients who have advanced or complicated TB. This creates more work for the already stretched TB clinic doctors who must coordinate care and consult other specialties over the phone for such patients. One caregiver shared that they had to rush their sick child to another hospital as hospital A was unable to admit and care for them inpatient.

Healthcare providers also shared that the unavailability of equipment, such as malfunctioning of X-ray machines, CT scan and MRI machines, can delay TB diagnosis or treatment. In addition, as these scans are expensive, most patients cannot afford to get them done at other facilities. There is often a long wait to get ultrasounds performed, which also causes delays in diagnosis.

“X-ray machine—this is the sixth time it has shut down since I have joined. Our OPD is running without an X-ray machine, and we cannot refer the patients to Hospital A’s main X-ray room because they can spread TB. Sending patients home without an X-ray and diagnosis often means patients not coming back and starting treatment.” (Healthcare provider ID 103)

7.Patient–provider relationships

Caregivers and healthcare providers discussed the positive attitude of Hospital A’s staff, and how they take ownership of their work. This leads to satisfaction of the patients and their families, which in turn motivates them to overcome initial hurdles and have their child diagnosed and started on treatment on time.

“The way they took care of my child I knew I was at the right place.”(Caregiver ID 93, female)

“Although it is difficult for us because we are also overworked, but we have to make time—we do not compromise on the counselling. I counsel my patients myself. But if there is anything that I have missed, the counsellors cover that.”(Healthcare provider ID 105)

## 5. Discussion

We found that the length of time required to be diagnosed and treated (~3 months) for children and adolescents in Karachi is longer than what has been reported in many studies of adults with TB in South Asia [12], and the major barriers driving delays were related to financial constraints of caregivers and involvement of general physicians. Almost all caregivers who were interviewed mentioned financial constraints as the main barrier to their child’s diagnosis, and these constraints were exacerbated by social factors that prevented women from taking children to the doctor. Moreover, in our study, 69% of caregivers first sought care with GPs, but they felt that the GPs were unable to provide satisfactory care. Many drivers of delay that we identified, such as financial constraints and distance from health facilities affect entire families, are commonly reported in studies of adults. Some factors, like gender norms, had specific manifestations that related to the health care of children in the family [19]. This shows that the failure to promptly diagnose children with TB is not simply a result of clinical complexity, but a manifestation of the many barriers to accessing healthcare affecting impoverished communities.

Our finding that costs of seeking treatment and distance to health facilities were barriers to prompt diagnosis of children and adolescents with TB has been observed in diverse settings [13,20]. Providers in our study mentioned that systematic mass community screening efforts are useful for reducing delays in diagnosis. Indeed, community-based interventions that actively seek out and identify people at risk for TB are known to increase TB diagnoses among adults [21] and reduce costs incurred by patients in the process of accessing diagnosis and treatment [22,23]. However, for children and younger adolescents, it is important that community-based screening be coupled with strengthening the ability of health services to diagnose pediatric TB; otherwise screening alone can be ineffective [24].

The high utilization of private general physicians in our study is consistent with the large role of the private sector in providing healthcare in much of South Asia [25]. In research done in Pune, India in 2016, 74% of children had visited private healthcare providers (often multiple times) before accessing the public sector [14]. In Pakistan, children and adolescents who present to GPs with symptoms suggestive of TB tend to go undiagnosed as TB is not considered a common illness and is difficult to diagnose in younger age groups [26]. Training private GPs in pediatric TB diagnosis has been shown to be effective in addressing this issue. In a study done in a rural setting of Pakistan in 2014–2016, the number of child TB cases increased three-fold partly due to training and incentivization of GPs during the study period [27]. Therefore, GP training programs focused on child and adolescent TB should be made a part of the government strategy to eliminate TB.

The study had a few limitations. Our quantitative survey relied on caregiver recall, which was a challenge if there were many healthcare visits or if the length of time between onset of symptoms and treatment initiation was long. However, among the subset of caregivers who were interviewed, information collected during the survey tended to be consistent with what was said during the interview, giving us confidence in our survey results. Another limitation is that the study was conducted at only one private hospital in which all care is free, so patient experiences within the hospital and findings about hospital processes are not necessarily generalizable to other health care settings. A larger study encompassing a range of health care settings and geographic locations, as well as larger sample sizes, could further strengthen our ability to understand the most important drivers of time to diagnosis. There were also limitations in our ability to comprehensively capture barriers. Younger age groups of children were poorly represented in the study due to small numbers of diagnoses, which could mean that we did not fully capture the barriers to diagnosis for this age group. Also, we interviewed caregivers since they are responsible for healthcare-seeking decisions and thus were best placed to answer our research question, but we did not capture the experience of the children or adolescents themselves through interviews. Finally, our survey did not ask about family income levels among family members, preventing us from comparing results across socioeconomic groups.

## 6. Conclusions

Our study adds to the global literature on delayed diagnosis of pediatric and adolescent TB, reporting longer times to diagnosis than prior studies from the region and identifying contributors to these delays that are distinct from those experienced by adults with TB. Our findings suggest that interventions to decentralize TB services for children and adolescents, such as community screenings, and to increase the capacity of primary-level GPs are required to reduce delays in diagnosis and treatment initiation for children and adolescents. The coverage and quality of community screenings should be continuously monitored, and there should be linkages established between GPs and pediatric TB experts to ensure continued learning and performance. In addition, more research is needed to identify optimal approaches to reduce barriers to timely child and adolescent TB diagnosis in different settings and the impact of those approaches.

## Figures and Tables

**Figure 1 ijerph-22-01477-f001:**
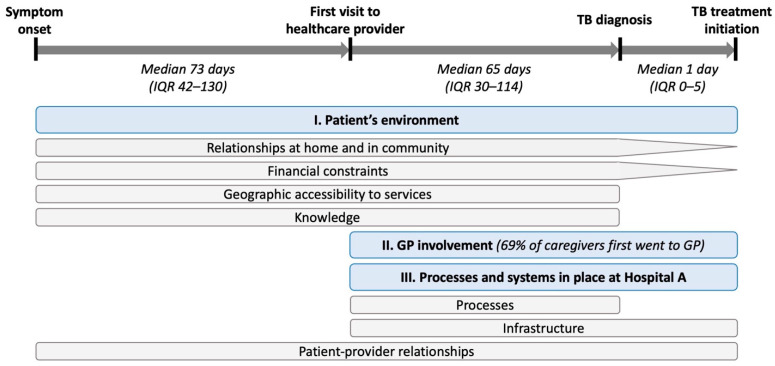
Joint display showing the journey from symptom onset to treatment initiation. Quantitative measures are shown in italic text. Barriers and facilitators to completing each stage of the journey identified from qualitative data are shown in boxes under the stages of the journey that they impact. Blue boxes represent major qualitative thematic categories, while grey boxes represent sub-categories. Abbreviations: IQR = interquartile range, GP = general practitioner.

**Table 1 ijerph-22-01477-t001:** Characteristics of pediatric TB patients whose caregivers were surveyed (N = 100) and interviewed (N = 30) at a hospital in Karachi in 2023.

Characteristics of Participants		Survey n (%)	Interview n (%)
Sex	Male	23 (23)	13 (43)
	Female	77 (77)	17 (57)
Age group	0–4 years	7 (7)	5 (17)
	5–9 years	11 (11)	5 (17)
	10–14 years	43 (43)	12 (40)
	15–18 years	39 (39)	8 (27)
Relation of caregiver	Parent	80 (80)	27 (90)
	Sibling	11 (11)	1 (3)
	Other	9 (9)	2 (7)
Number of children in home	1–3	30 (30)	15 (50)
	4–6	54 (54)	9 (30)
	>6	16 (16)	6 (20)
Household contact with TB	Within 2 years	19 (19)	8 (27)
	Over 2 years ago	40 (40)	10 (33)
	None	41 (41)	12 (40)
Type of TB	Pulmonary	58 (58)	17 (57)
	Extrapulmonary	36 (36)	12 (40)
	Pulmonary and extrapulmonary	6 (6)	1 (3)
Bacteriologic confirmation	Yes	63 (63)	21 (70)
	No	37 (37)	9 (30)
Drug sensitivity (confirmed or presumed *)	Drug-sensitive	92 (92)	23 (77)
	Drug-resistant	8 (8)	7 (23)

* Confirmed drug sensitivity is based on drug susceptibility testing of bacteriologic samples. In the absence of bacteriologic confirmation, drug sensitivity is presumed if the child/adolescent is a contact of a person with drug-resistant TB.

**Table 2 ijerph-22-01477-t002:** Time to treatment and healthcare visits among pediatric TB patients whose caregivers were surveyed (N = 100).

	Median	IQR
Interval from symptom onset until treatment (in days)	91	58–160
Days from symptom onset until first healthcare visit	73	42–130
Days from first visit until TB diagnosis	65	30–114
Days from diagnosis until treatment initiation	1	0–5
Number of total healthcare visits	4	3–6
Healthcare visits to private practice	1	1–2
Healthcare visits to spiritual healer	0	0–1
Healthcare visits to pharmacy	1	1–2
Healthcare visits to lab	2	1–2
Healthcare visits to hospital	3	2–4

## Data Availability

The data presented in the study are available from the corresponding author upon reasonable request.

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
