# Peer review of "Barriers and Facilitators to Timely Diagnosis of Tuberculosis in Children and Adolescents in Karachi, Pakistan"

_ijerph, 2025, doi:10.3390/ijerph22101477_

Round 1

Reviewer 1 Report

Comments and Suggestions for Authors

Dear Editor,

Thank you for the opportunity to read this interesting study.  I would welcome the wider presentation of the results of this study,, but have some concerns relating to the manuscript in its present form.  

There is a deal of well conducted research here and the results are of interest to a wider audience, but my concerns are:

There is no mention in this manuscript that the study received ethical approval. Publication cannot proceed without this.

The study instruments were translated into the local language (Urdu).  The gold-standard for international use of study instruments in other languages/cultures is full independent validation of the instrument in the language and culture where it is to be used.  In the absence of this, at least, independent forward and back translation are required, to ensure the meaning is accurately reflected.  The manuscript only mentions one-way translation, and no validation of readability and comprehensibility in the target population.  Checking against international standards by highly educated professionals is unreliable as their reading age and vocabulary is likely to be higher than the target population.

The methods are well described, subject to the comments above about the study instruments.  If forward and back translation was not applied this needs to be clearly stated so that readers can consider the likely effects on comprehension of the survey instruments by participants.

I would like the authors to specify whether only patients with pulmonary tuberculosis are diagnosed and treated in this private hospital or are they also equipped to diagnose extrapulmonary tuberculosis. Have there been any cases of tuberculous meningitis in this clinic? What was the interval between the time of diagnosis and the initiation of antituberculosis medication? Did the delay in diagnosis lead to an increase in mortality and sequelae?

I suggest introducing the following literature articles [https://doi.org/10.1515/rrlm-2015-0016 and Clinical aspects of tuberculous meningitis in children. Revista Medico-chirurgicala a Societatii de Medici si Naturalisti din Iasi. 2010 Jul-Sep;114(3):743-747. PMID: 21243801] which can provide useful insights into the diagnosis of tuberculosis in children and adolescents.

It would have been interesting if the authors had made a correlation between the number of days from the first consultation to the initiation of antituberculosis treatment and the mortality and sequelae rate.

Were there any cases of TB-HIV coinfections diagnosed in this clinic? If so, did these cases pose more problems in terms of monitoring, were there any cases of therapeutic abandonment?

The results are clearly presented.  

The tables and figures are clear and do not require modifications or clarifications.

The discussion is appropriate and the authors specify what the limitations of this study.

The conclusions section could be better structured and presented separately.

Comments on the Quality of English Language

The english language requires minor modifications.

Author Response

Comment 1: There is no mention in this manuscript that the study received ethical approval. Publication cannot proceed without this.

Response: The details of ethical approval are included under the “Institutional review board statement” header in the Declarations section, as required by the journal.

Comment 2: The study instruments were translated into the local language (Urdu).  The gold-standard for international use of study instruments in other languages/cultures is full independent validation of the instrument in the language and culture where it is to be used.  In the absence of this, at least, independent forward and back translation are required, to ensure the meaning is accurately reflected.  The manuscript only mentions one-way translation, and no validation of readability and comprehensibility in the target population. Checking against international standards by highly educated professionals is unreliable as their reading age and vocabulary is likely to be higher than the target population.

Response: We do not state that study instruments were translated into the local language. They were developed and administered verbally in Urdu by the lead author, a native Urdu speaker who is fully bilingual in Urdu and English.  Data collection did not require participants to read anything, nor did it require standardization of language across data collectors.  Translation/back-translation of instruments is therefore not an issue.  We have attempted to make this clearer by adding the bolded phrase to the sentence: “The lead author, a native Urdu speaker, verbally administered a 30-minute survey to caregivers at the TB clinic or over the phone in Urdu.”

Comment 3: The methods are well described, subject to the comments above about the study instruments.  If forward and back translation was not applied this needs to be clearly stated so that readers can consider the likely effects on comprehension of the survey instruments by participants.

Response: Please see the response to the comment above.

Comment 4: I would like the authors to specify whether only patients with pulmonary tuberculosis are diagnosed and treated in this private hospital or are they also equipped to diagnose extrapulmonary tuberculosis. Have there been any cases of tuberculous meningitis in this clinic? What was the interval between the time of diagnosis and the initiation of antituberculosis medication? Did the delay in diagnosis lead to an increase in mortality and sequelae?

Response: We have added the sentence “All forms of TB, including drug-resistant TB, extrapulmonary TB, and TB meningitis are diagnosed and treated” to the description of the hospital in the Methods.  The study data included 3 TB meningitis cases. In all 3 cases, the interval between the time of diagnosis and treatment initiation were 0 days. However, the interval between first visit to any healthcare provider to the time of diagnosis were 92 days (the only drug-resistant case for TB meningitis in the data), 58 days, and 35 days respectively. Clinical outcomes such as mortality and sequelae were beyond the scope of the study, and we did not collect data on these.

Comment 5: I suggest introducing the following literature articles [https://doi.org/10.1515/rrlm-2015-0016 and Clinical aspects of tuberculous meningitis in children. Revista Medico-chirurgicala a Societatii de Medici si Naturalisti din Iasi. 2010 Jul-Sep;114(3):743-747. PMID: 21243801] which can provide useful insights into the diagnosis of tuberculosis in children and adolescents.

Response: The first DOI link does not appear to correspond to a document. The second is a 15-year old article in Romanian that we are unable to read. While the abstract suggests that there may be data quantifying delays in diagnosis of TB meningitis in the article, the fact that the findings are somewhat dated, focused on a single form of TB, and from a single hospital in a high-income country make it less relevant for the discussion of our study.

Comment 6: It would have been interesting if the authors had made a correlation between the number of days from the first consultation to the initiation of antituberculosis treatment and the mortality and sequelae rate.

Response: Thank you for the suggestion. Unfortunately, we didn't collect data on what happened after TB treatment initiation for this study, as it was not directly relevant to the study objective of understanding delays in diagnosis. However, we will keep this in mind for any future work conducted on the topic.

Comment 7: Were there any cases of TB-HIV coinfections diagnosed in this clinic? If so, did these cases pose more problems in terms of monitoring, were there any cases of therapeutic abandonment? The results are clearly presented.  

Response: Yes, there are TB-HIV coinfections diagnosed in this clinic, but we did not collect data related to clinical management as this was not directly relevant to the study objective of understanding delays in diagnosis.

Comment 8: The tables and figures are clear and do not require modifications or clarifications.

Response: Okay, thank you

Comment 9: The discussion is appropriate and the authors specify what the limitations of this study.

Response: Okay, thank you

Comment 10: The conclusions section could be better structured and presented separately.

Response: We have separated the conclusions section as suggested. We have also added a sentence to the beginning of the conclusion (“Our study adds to the global literature on delayed diagnosis of pediatric and adolescent TB, reporting longer times to diagnosis than prior studies from the region and identifying contributors to these delays that are distinct from those experienced by adults with TB.”) to help set it apart from the rest of the discussion and transition to our recommendations.

Reviewer 2 Report

Comments and Suggestions for Authors

Thank you for the opportunity to review this important and timely study addressing barriers and facilitators to the timely diagnosis of tuberculosis. The research is of significant interest and offers valuable insights; indeed, its impact could be further strengthened by replicating the study with a larger sample size and including multiple cities and healthcare facilities.

Introduction:
It is recommended that the introduction include epidemiological figures concerning the prevalence of tuberculosis both nationally (in Pakistan) and specifically within the study region. Presenting these data will help contextualize the importance of the research.

Objectives:
The objectives of the study as currently presented are not sufficiently clear and should be elaborated to provide better direction and clarity to readers.

Setting:
In the description of the study setting, it would be beneficial to specify the total annual number of TB patients managed by the hospital, which would provide context for the scale and representativeness of the site.

Results (Section 4.2.1, Patient’s Environment):
Some qualitative findings are presented in a manner that resembles discussion or interpretation rather than reporting direct study results. It is advisable to distinguish the main findings more clearly from their interpretation.

Socioeconomic Data:
The study did not inquire about participants’ income levels, which limits the ability to analyze results across different socioeconomic groups. Inclusion of such data is recommended for future research.

Sample Size and Generalizability:
Given the large and diverse population of Pakistan, the sample size in both the qualitative and quantitative components is relatively small. As such, the generalizability of findings should be interpreted with caution. The manuscript appropriately notes some limitations, which partially address this concern.

Household TB History:
It would also be informative to know whether caregivers were asked about recent or past tuberculosis diagnoses among other family members, as this could have an important bearing on health-seeking behavior and diagnostic timelines.

Overall, the study is promising and contributes valuable knowledge, but addressing these points would enhance its rigor and reader impact.

Author Response

Comment 1: The research is of significant interest and offers valuable insights; indeed, its impact could be further strengthened by replicating the study with a larger sample size and including multiple cities and healthcare facilities.

Response: Thank you for the suggestion. While we already acknowledge the limitations of the single study site in the discussion, we have added another sentence emphasizing the value of replication with a larger sample size and multiple sites: “A larger study encompassing a range of health care settings and geographic locations, as well as larger sample sizes, could further strengthen our ability to understand the most important drivers of time to diagnosis.”

Comment 2 (Introduction): It is recommended that the introduction include epidemiological figures concerning the prevalence of tuberculosis both nationally (in Pakistan) and specifically within the study region. Presenting these data will help contextualize the importance of the research.

Response: We have added the following sentence and related citations to the introduction: “Pakistan has the fifth highest TB burden globally with an estimated incidence of 277 per 100,000 people in 2023, and the Sindh region where the city of Karachi is located is estimated to have the highest TB incidence in the country.”

Comment 3 (Objectives): The objectives of the study as currently presented are not sufficiently clear and should be elaborated to provide better direction and clarity to readers.

Response: Thank you, the editor also noted this issue and suggested a rephrasing of the statement of objective.  We have revised the statement as the editor suggested: “We therefore conducted the present study to identify barriers and facilitators to timely diagnosis of TB in children and adolescents.  This will both add to the global literature and inform national policies for enhancing the care of TB-affected children and adolescents.”

Comment 4 (Setting): In the description of the study setting, it would be beneficial to specify the total annual number of TB patients managed by the hospital, which would provide context for the scale and representativeness of the site.

Response: This sentence has been added to the study setting description: “Additionally, the hospital’s pediatric TB program registers 150 to 200 children and adolescents with TB every month, who are often referred with symptoms suggestive of TB from other health facilities”.

Comment 5 (Results - Section 4.2.1, Patient’s Environment): Some qualitative findings are presented in a manner that resembles discussion or interpretation rather than reporting direct study results. It is advisable to distinguish the main findings more clearly from their interpretation.

Response: We acknowledge that the language we used did not always make it clear that we are reporting things stated by participants and not our interpretation of these statements. We have reviewed the qualitative results to clarify this language and ensure that it is clear that we are reporting rather than interpreting.

Comment 6 (Socioeconomic Data): The study did not inquire about participants’ income levels, which limits the ability to analyze results across different socioeconomic groups. Inclusion of such data is recommended for future research.

Response: We have added the following sentence to the limitations: “Finally, our survey did not ask about family income levels among family members, preventing us from comparing results across socioeconomic groups.”

Comment 7 (Sample Size and Generalizability): Given the large and diverse population of Pakistan, the sample size in both the qualitative and quantitative components is relatively small. As such, the generalizability of findings should be interpreted with caution. The manuscript appropriately notes some limitations, which partially address this concern.

Response: We agree, and we have strengthened the statement of limitations by adding the following sentence to the statement about limited generalizability: “A larger study encompassing a range of health care settings and geographic locations, as well as larger sample sizes, could further strengthen our ability to understand the most important drivers of time to diagnosis.”

Comment 8 (Household TB History): It would also be informative to know whether caregivers were asked about recent or past tuberculosis diagnoses among other family members, as this could have an important bearing on health-seeking behavior and diagnostic timelines.

Response: We have added this information to Table 1.

Reviewer 3 Report

Comments and Suggestions for Authors

Dear Authors,

I have reviewed your article entitled “Barriers and facilitators to timely diagnosis of tuberculosis in children and adolescents in Karachi, Pakistan”.

As part of the program to end TB, identifying barriers to diagnosis and treatment and implementing measures to address them is crucial. Identifying these barriers is especially crucial in countries with high TB ​​prevalence. Therefore, your work is very valuable. Thank you for your efforts.

Overall, it's well designed and written. You should just make a few revisions:

  • WHO website (https://www.who.int/health-topics/tuberculosis#tab=tab_1) should also be cited as a reference along with the first reference.
  • 'NGO' s explanation should be included in 1. Setting.
  • Although there are 63 patients with bacteriologic confirmation in Table 1, drug sensitivity was determined in 92 patients. Although the presumed is stated, the presumed should be explained, or only those with drug sensitivity in the bacteriologic confirmation should be stated. Furthermore, the drug sensitivity section in Table 1 remained as 22 + 7 in the interview. This should be corrected.
  • In Table 2, in the text, and in Figure 1, I think "mean" should be used instead of "median."

Best regards,

Author Response

Comment 1: WHO website (https://www.who.int/health-topics/tuberculosis#tab=tab_1) should also be cited as a reference along with the first reference.

Response: We have added a citation to the most recent WHO Global TB report along with the first reference.  We refrain from citing websites, which change frequently and can cause problems with inactive links. The annual TB reports are archived by WHO and are always accessible, and contain similar information as what is on the website.

Comment 2: 'NGO' s explanation should be included in 1. Setting.

Response: We have replaced the acronym with “non-governmental organizations”

Comment 3: Although there are 63 patients with bacteriologic confirmation in Table 1, drug sensitivity was determined in 92 patients. Although the presumed is stated, the presumed should be explained, or only those with drug sensitivity in the bacteriologic confirmation should be stated. Furthermore, the drug sensitivity section in Table 1 remained as 22 + 7 in the interview. This should be corrected.

Response: We have revised the table to clarify that drug sensitivity is either confirmed or presumed, and we have added a footnote to explain: “Confirmed drug sensitivity is based on drug susceptibility testing of bacteriologic samples. In the absence of bacteriologic confirmation, drug sensitivity is presumed if the child/adolescent is a contact of a person with drug-resistant TB.”  We have corrected the number of interviewed patients in the drug sensitivity section of Table 1 (it should be 23, not 22 drug-sensitive).

Comment 4: In Table 2, in the text, and in Figure 1, I think "mean" should be used instead of "median."

Response: The data are not normally distributed (they are right-skewed with high-value outliers), which makes the median rather than the mean a better measure of central tendency.

Round 2

Reviewer 1 Report

Comments and Suggestions for Authors

The manuscript was revised according to the reviewers' instructions and was significantly improved.

It may be published in its current form.